# Diagnostic Validation of a Comprehensive Targeted Panel for Broad Mutational and Biomarker Analysis in Solid Tumors

**DOI:** 10.3390/cancers14102457

**Published:** 2022-05-16

**Authors:** Guy Froyen, Ellen Geerdens, Severine Berden, Bert Cruys, Brigitte Maes

**Affiliations:** Laboratory of Molecular Diagnostics, Department of Clinical Biology, Jessa Hospital, B-3500 Hasselt, Belgium; ellen.geerdens@jessazh.be (E.G.); severine.berden@jessazh.be (S.B.); bertcruys@hotmail.com (B.C.); brigitte.maes@jessazh.be (B.M.)

**Keywords:** Next Generation Sequencing (NGS), precision medicine, targeted screening, TSO500, validation

## Abstract

**Simple Summary:**

The analysis of tumor-associated genetic variants and biomarkers is critical for therapy choice, as specific mutations allow for a personalized treatment. Because more and more mutation-treatment combinations become available, screening should be performed on many genes simultaneously. The use of large and comprehensive gene panel screenings in molecular diagnostics, however, requires an extensive and thorough validation to demonstrate the correctness of all clinically relevant data. Here, we describe such validation using a large number of samples and confirmed effective detection of several types of mutations for different validation parameters. Samples of tumor patients thus can be reliably tested with a comprehensive assay to maximize their personalized treatment regimen.

**Abstract:**

The use of targeted Next Generation Sequencing (NGS) for the diagnostic screening of somatic variants in solid tumor samples has proven its high clinical value. Because of the large number of ongoing clinical trials for a multitude of variants in a growing number of genes, as well as the detection of proven and emerging pan-cancer biomarkers including microsatellite instability (MSI) and tumor mutation burden (TMB), the currently employed diagnostic gene panels will become vastly insufficient in the near future. Here, we describe the validation and implementation of the hybrid capture-based comprehensive TruSight Oncology (TSO500) assay that is able to detect single-nucleotide variants (SNVs) and subtle deletions and insertions (indels) in 523 tumor-associated genes, copy-number variants (CNVs) of 69 genes, fusions with 55 cancer driver genes, and MSI and TMB. Extensive validation of the TSO500 assay was performed on DNA or RNA from 170 clinical samples with neoplastic content down to 10%, using multiple tissue and specimen types. Starting with 80 ng DNA and 40 ng RNA extracted from formalin-fixed and paraffine-embedded (FFPE) samples revealed a precision and accuracy >99% for all variant types. The analytical sensitivity and specificity were at least 99% for SNVs, indels, CNVs, MSI, and gene rearrangements. For TMB, only values around the threshold could yield a deviating outcome. The limit-of-detection for SNVs and indels was well below the set threshold of 5% variant allele frequency (VAF). This validated comprehensive genomic profiling assay was then used to screen 624 diagnostic samples, and its success rate for adoption in a clinical diagnostic setting of broad solid tumor screening was assessed on this cohort.

## 1. Introduction

The detection of particular gene variants in a specific tumor type and stage has been increasingly associated with personalized medicine. The number of clinically relevant variants including targeted treatment sensitivity as well as resistance-inducing variants is currently highest in advanced stages of non-small cell lung carcinoma (NSCLC). However, in the last few years, an increasing number of hotspot variants has been associated with diagnostic, prognostic, or therapeutic value in several other tumor types. Therefore, in most diagnostic laboratories, the single gene assays have steadily been replaced by multi-gene analyses using Next Generation Sequencing (NGS) for that purpose [1].

Initially, only SNVs and indels could be detected by the smaller gene panels traditionally harboring hotspots in only a handful of genes. A few years ago, larger panels consisting of hotspots in up to 50 genes generated not only more uniform coverages that ameliorated the robustness of the assay but also allowed the detection of CNVs for those genes with sufficient amplicons in the panel. Clinically relevant CNVs include *ERBB2* amplification in breast cancer and *EGFR* and *MET* amplifications as a resistant mechanism in non-small cell lung carcinoma (NSCLC). Moreover, the simultaneous analysis of RNA extracted from the same samples allowed not only for the screening of fusion genes in specific tumor types (e.g., *ALK* and *ROS1* fusions in NSCLC) but also in a tumor type agnostic setting (e.g., *NTRK* fusions) [2]. In addition, RNA-based analysis could screen for clinically relevant exon skipping events in *EGFR, AR*, and *MET* [3]. The detection of fusion genes by NGS thus can replace the labor-intensive fluorescent in situ hybridization (FISH) assay for which multiple probes and tissue sections are required for all single relevant genomic rearrangements. Furthermore, in contrast to the mainly used single-gene break-apart FISH approach, NGS analysis also reveals the fusion partner.

Next to PD-L1 expression, MSI and TMB are biomarkers that have shown clinical utility in predicting response to immunotherapy [4,5]. Upon the inactivation of mismatch repair genes, DNA replication errors accumulate and result in a high number of neo-epitopes that stimulate the immune response. The increase in genomic variants can be measured by the analysis of the mutation rate of microsatellites (MSI) or the overall number of variants in the coding domain of the genome (TMB), expressed as the number of variants per interrogated sequence content (in Mb). While MSI assessment can also be achieved with smaller panels, only large panels with a content of at least 1.5 Mb can correctly calculate the TMB value [6,7]. In conclusion, comprehensive gene panels containing more than about 400 genes, such as the TSO500 assay, can interrogate all possible variants and biomarkers to assess the genetic make-up of a tumor for a maximal clinical value.

The personalized treatment options for patients carrying a solid tumor are steadily growing. Until recently, activating mutations in a restricted number of genes were amenable to precision medicine, but this number has increased significantly. Moreover, thousands of clinical trials are currently being conducted worldwide to investigate the clinical effect of promising novel drugs in response to specific (combinations of) variants in specific tumor types or even pan-cancerously [8]. Furthermore, existing drugs are now being tested in other tumor types harboring the same genetic variant ‘basket trials’, and multiple targeted therapies are evaluated for a single disease ‘umbrella trials’ [9,10]. These trials often select patients based on the occurrence of different variant types (e.g., SNVs, CNVs, fusions) or of biomarkers MSI and TMB, alone or in combination. Taken together, these studies will likely result in a large expansion of the number of genes or loci that need to be screened for personalized therapy in the near future. 

Analytical and clinical validation of the smaller NGS panels already is a tedious task since multiple performance characteristics need to be validated for each possible condition in a diagnostic laboratory setting. The validation of comprehensive panels is even more challenging because of the many different genes, variant types, biomarkers, tumors, and specimen types with different tumor content (TC) and variable DNA and RNA quantities and qualities. Here, we report on the diagnostic validation of a comprehensive genomic profiling assay using 170 retro- and prospective solid tumor samples, on DNA and RNA, for performance characteristics precision (repeatability and reproducibility), accuracy, limit-of-detection, analytical sensitivity, and specificity. Finally, we provide technical and molecular data on the screening of this validated assay on more than 600 routine diagnostic samples.

## 2. Materials and Methods

### 2.1. Samples and Orthogonal Assays

A total of 170 FFPE tumor samples of several different tumor and specimen types were selected (Table 1). The TC varied between 10% and 90%, and a large variety of sample tissue types was included (Table 2). All DNA samples had been analyzed previously with the accredited TruSeq Custom Amplicon assay (TSCA, Illumina, San Diego, CA, USA) as previously described [11] or the accredited AmpliSeq for Illumina assay (Illumina) with the AmpliSeq Focus panel supplemented with custom primers for SNV and indel analysis in 31 genes or for analysis of amplifications of the *EGFR*, *ERBB2*, and *MET* genes. The custom panel additionally included biomarkers for diagnostic MSI analysis. The Idylla MSI assay (Biocartis, Mechelen, Belgium) was used for additional confirmation. For SNV and indel analysis in *BRCA1* and *BRCA2*, the AmpliSeq for Illumina BRCA panel was used. For gene fusion detection and for exon skipping events in *EGFR* and *MET*, the data were compared with those obtained with the accredited AmpliSeq for Illumina Focus RNA panel (Illumina), and/or with Fluorescent in situ Hybridization (FISH). DNA and RNA extraction was performed on a Maxwell (Promega, Madison, WI, USA) with the FFPE DNA and RNA extraction kits, respectively, as described previously [11]. DNA and RNA concentrations were quantified on a Qubit fluorimeter 3.0 (Lifetechnologies, Gaithersburg, MD, USA) using the Qubit dsDNA Broad Range and RNA High Sensitivity assay kits, respectively. Samples were immediately processed or stored at −20 °C. For long-term storage, RNA was frozen at −80 °C. TSO500 analysis was performed mostly on stored DNA and RNA from fresh FFPE blocks. No material from archival tissue samples >2 years was used.

The study was conducted in accordance with the Declaration of Helsinki. The committee of Medical Ethics of the Jessa Hospital (Hasselt, Belgium) has approved this study. The protocol code is 19.53/klin19.01

As reference material for DNA analysis, we used the Structural Multiplex Reference Standard HD753 (HorizonDx) that harbors 10 confirmed variants (7 SNVs and 3 indels) at defined VAFs from 2.5% to 18.2%. Moreover, it includes CNVs in *MET* and *MYC-N* with 4.5 and 9.5 copies, respectively. On the RNA level, we used the SeraSeq FFPE Tumor Fusion RNA Reference material v2 (SeraCare) harboring 14 well-known fusions and 2 exon skipping events in *EGFR* (vIII) and *MET* (ex14 skipping).

### 2.2. TSO500 Library Preparation

Library preparation was performed according to the TruSight Oncology 500 Reference guide (TruSight Oncology 500 Reference Guide (1000000067621)). Briefly, up to 80 ng DNA was sonicated with a Covaris ME220 (Covaris, Woburn, MA, USA) to fragments with a mean size of about 180 bp. After end-repair and A-tailing, ligation of adapters carrying the Unique Molecular Identifiers (UMIs) was performed and fragments were amplified to add the indexes. Next, hybridization was performed overnight followed by a streptavidin magnetic bead-based capture to enrich for the selected targets. A second 2 h hybridization and capture round was performed with the same probes, after which the enriched fragments were amplified in a second PCR step to produce the enriched library. Purified libraries were then bead-based normalized. For fusion detection, first and second strand cDNA synthesis was performed starting from up to 40 ng RNA, after which end-repair and A-tailing was conducted. The next steps were highly similar to those for DNA, resulting in the normalized enriched RNA-based libraries. Finally, the DNA- and RNA-based libraries were pooled, denatured, and diluted for instant sequencing. Individual and pooled libraries were stored at −20 °C.

### 2.3. NGS and Primary Data Analysis

The Pooled Amplicon Libraries from 8 DNA and 8 RNA libraries were loaded on a NextSeq 500/550 High Output Kit v2.5 (300 Cycles) and paired-end (2 × 101 bp) sequenced on a NextSeq500 instrument (Illumina). Data were analyzed with the TSO500 Local App v2.0 (Illumina) according to the user guide (TruSight Oncology 500 v2.0 Local App User Guide (1000000095997) (illumina.com; accessed on 5 May 2022)). It is important to note that deduplication is performed based on UMIs to remove all duplicate reads. Therefore, the indicated coverages are based on the unique reads only. DNA and RNA degradation was checked by calculating the mean amplicon insert size and deamination by analysis of the frequency of C > T; G > A changes. TMB was calculated as the total number of somatic non-hotspot variants with VAF > 5% per Mb of interrogated sequence. For MSI analysis, at least 40 MSI sites need to be interrogated. All passed-filter data including TMB, MSI, gene amplifications, splice variants, fusions, and small variants (SNVs and indels) are provided in the CombinedVariantOutput.tsv file of each sample. Only these data were considered for validation. 

### 2.4. Variant Analysis

Variant annotation and filtering were performed in VariantStudio (Illumina) on the PASS filter variants only and sequencing reads from the Bam files were visualized in Integrated Genome Viewer (IGV; The Broad Institute). Variant classification and interpretation was performed according to the two-level-based Belgian ComPerMed guidelines [12], i.e., tumor type independent biological classification followed by its clinical interpretation (Guidelines_april_2021-NL.pdf (compermed.be)). Briefly, for the biological classification, all variants that are not regarded as technical artifacts are first checked for their presence in GnomAD (v2.1.1). Those with a minor allele frequency (MAF) >0.1% or >1% in at least one ethnic population are classified as Likely Benign or Benign, respectively. The remaining variants are judged as non-polymorphic and further checked at the amino acid level for their presence in the Consensus Pathogenic Variants (CPV) list, which harbors all amino acid changes in tumor-related proteins that are clear-cut drivers (hotspot variants). Variants present in this CPV list are classified as Pathogenic. The remaining variants can then only be classified as Likely Pathogenic or Variant of Unknown Significance (VUS). Those resulting in a clear loss-of-function (LoF; i.e., stop, frameshift, …) in a tumor suppressor gene are classified as Likely Pathogenic, while LoF variants in oncogenes end up as VUS. Non-LoF variants (missense, inframe indels, …), finally, need to be classified according to a Scoring Table that requires variant checking in several cancer-related databases, publications, prediction tools, etc. [12]. As the clinical interpretation is based on the biological classification, only Pathogenic and Likely Pathogenic variants are clinically interpreted on the report and the ACMG/AMP/CAP tiering system is used [13].

## 3. Results

### 3.1. Quality Metrics

For TSO500 analysis, 80 ng DNA and 40 ng RNA are recommended as input material. However, for 7/90 (7.7%) DNA and 10/80 (12.5%) RNA samples (of which there was one no-RNA control) these minimal amounts were not obtained. Additionally, a library concentration (before normalization) of at least 10 ng/µL is advised, which was the case for all DNA and 93.1% of the RNA validation samples. The mean yield (±SD) per run for eight DNA and eight RNA samples was 98 (±13) gigabases. For reliable data, the median insert size at the DNA level should be >70 bp and the median target coverage >150×, which was the case for 99.13% and 95.65% of the validation samples, respectively. At the RNA level, the median insert size should be >80 bp and the total on target reads >9,000,000. These thresholds were met in 100% and 93.25% of the validation samples, respectively. 

### 3.2. Precision

The reproducibility was first tested on the HD753 reference DNA control sample in 10 different sequencing runs. Following the ComPerMed guidelines, this reference sample carried 10 Pathogenic and 26 Likely Pathogenic variants (17 SNVs and 19 indels) with VAFs ranging from 3.2% to 21.3%. In each run, all variants were detected with highly comparable VAFs with SDs < 2.7% (Appendix A). Moreover, the copy number of both amplifications (*MET* and *MYC-N*) as well as the MSI (mean 70% ± 3%) and TMB (mean 250 ± 5) values from this reference were also very similar, demonstrating the high reproducibility of the assay for all variant types and biomarkers using this artificial sample. Next, the intrarun (repeatability) and interrun (reproducibility) variability were tested on three retrospective diagnostic DNA samples to assess the precision of diagnostically important SNVs and indels (Figure 1a), amplifications of *ERBB2* and *EGFR* (Figure 1b), and the MSI and TMB values. Two out of three samples (S1 and S3) were known to be MSI-high (MSI-H) and TMB-high (TMB-H) (Figure 1c,d). 

For each variant type or biomarker, we included both high and low values to mimic the broad range of potential results. The precision was maximal as all SNV, indel, and CNV variants were detected with highly similar VAFs or fold changes (FC). Additionally, biomarker values were also almost identical between repeats. Finally, diagnostic RNA samples were interrogated for detection of an exon skipping event as well as for gene fusions. All three structural aberrations were unequivocally called all with a significantly high number of supporting reads (Figure 1e).

### 3.3. Analytical Sensitivity and Specificity

We tested a cohort of 74 diagnostic DNA samples extracted from 26 retrospective and 48 prospective FFPE samples carrying a total of 68 subtle variants (62 SNVs and 6 indels; VAF 5–97%; mean 35%; SD 24%) in 19 genes, of which 47 variants were unique (Appendix A), and six amplifications of the diagnostically relevant *EGFR* (one sample), *ERBB2* (three samples), or *MET* (two samples) gene with fold changes between 2.7 and 9.1. Four MSI-H retrospective samples were included, and two additional prospective samples were MSI-H. All variants and MSI-H markers were correctly detected by TSO500 and no false positive variants or MSI discordant samples were identified compared to the data obtained with the accredited methods. For TMB assessment, the results of 15 samples that had been analyzed with the FoundationOne assay (FoundationMedicine) were compared with those obtained with the TSO500 assay (Table 3). Setting the threshold for TMB-high (TMB-H) at 10 variants/Mb, eight samples were TMB-H in both assays, of which six were concordant as TMB-H. Two samples yielded a different TMB class. Sample t4 had 6 variants/Mb with FoundationOne, while this was 10 with TSO500 analysis. For sample t6, these values were 10 and 9, respectively, which is a borderline discordancy. Clear-cut TMB-low or -high samples were always correct, while samples with values close to the threshold of 10 variants per Mb are inherently more prone to be classified differently. Finally, for fusion gene and exon skipping events, 25 retrospective, 54 prospective, and the SeraSeq FFPE RNA reference sample were tested for concordance with data of the accredited AmpliSeq assay. These samples carried a total of 35 known fusions with 12 driver genes and 5 splicing events affecting *EGFR* or *MET*. All structural variants were confirmed. Of these, 24 were unique events (Appendix A). The number of fusion-supporting reads from the TSO500 analysis ranged from 24 to 6885 (mean 1899; SD 2054). Next to the confirmed fusion genes, five unknown fusions were found in the 54 prospective samples, but these could not be confirmed due to the unavailability of an alternative accredited assay. Of these, only the SLC45A3::ETV1 is a known fusion and also has a high number of supporting reads. The other fusions might be aspecific rearrangements or false positives, especially those supported by less than 20 reads. The remaining 49 samples in which no fusion or exon skipping event was detected were also found to be negative by the orthogonal assay.

Therefore, the sensitivity as well as specificity of the TSO500 assay for the SNVs, indels, CNVs, and fusions, as well as that of the MSI biomarker, were >99%. TMB scoring in samples with TMB values around the threshold is inherently prone to some variability. 

### 3.4. Limit of Detection (LoD)

For LoD analysis, dilution series were made by mixing samples with known variants, gene aberrations, or biomarkers. For each LoD assay, two samples were mixed at variable mix percentages. First, we selected 10 samples with a total of 24 variants (16 SNVs and 8 indels) at VAF 7% to 92% and with a tumor content (TC) between 30% and 90%. Two samples were mixed in such a way that they contributed 75%, 50%, and 25%, or 85%, 50%, and 15% of the original sample in the resulting mixture. All variants (22) that were tested down to the 15% or 25% contribution were detected (Appendix A). Of those, 13 (8 SNVs and 5 indels) had a VAF < 5%, clearly demonstrating that variants below that threshold can be efficiently detected as well. Nevertheless, we have set the variant-reporting threshold for SNVs and indels at a conservative VAF of 5%. 

For LoD analysis of CNVs, four samples with TC 30% to 60%, of which three harbored a clinically relevant CNV, were mixed two by two as described above. The original fold changes (FC), based on the AmpliSeq NGS assay, were 13.0 (*EGFR*; sample A1), 5.1 (*ERBB2*; sample A2), and 2.0 (*EGFR*; sample A3). Setting the FC threshold for a CNV at 1.8, we could detect the amplifications in samples A1 and A2 to the 25% contribution mixtures but not the *EGFR* CNV in sample A3 from the 50% contribution onwards (Appendix A). Interestingly, TSO500 also revealed a yet-undetected amplification at 11q13 in sample 2, including the genes *CCND1*, *FGF19*, *FGF4*, and *FGF3* with FCs gradually decreasing from 2.8–3.2 at the 75% mixture to 2.1–2.3 in the 50% mixture. At the contribution of 25%, the FCs fell below the threshold of 1.8. No CNV was found with either of both assays in the negative control sample A4.

The same procedure was performed for four samples, of which three carried a fusion in *RET*, *ALK*, or *ERG* with a highly variable number of supporting reads (126 to 2878 in undiluted samples). One sample without a fusion was used as negative control. Again, samples were mixed two by two to yield 75% and 25% contributions to the mixture. All fusions were detected in samples F1, F2, and F3 down to the 25% contribution supported by 16 to 322 fusion reads (Appendix A). No fusion was detected in the negative control sample F4. Finally, the SeraCare reference RNA sample was diluted 5 and 15 times in buffer (20% and 6.7% contribution, respectively) and analyzed. While all 14 gene fusions and 2 exon skipping events were detected with 8 to 174 supporting reads in the undiluted sample, a 5-fold dilution prohibited the detection of the three fusion/skipping events with the lowest number of supporting reads (8 to 32) in the undiluted sample. These events were not picked up by the software in the 5-times diluted mixture due to an insufficient number of supporting reads. A similar finding was obtained in the 15-times diluted sample in which still 10 rearrangements were correctly called, although with a low number (7 to 16) of supporting reads (Appendix A).

### 3.5. Accuracy

At the DNA level, the accuracy was tested on the reference control sample HD753 in 10 sequencing runs, on all retro- and prospective samples that were used for the sensitivity and specificity analysis, and on 5 External Quality Control (EQC) TMB samples. All 36 (Likely) Pathogenic variants in HD753 were detected with very low SDs as described previously (Appendix A), demonstrating a high accuracy. Moreover, all 68 known variants and CNVs in the retro- and prospective samples were detected as well. Analysis of the five EQC TMB samples with the FoundationOne-derived TMB scores of 10, 5, 14, 15, and 69 variants/Mb, respectively, yielded 5, 7, 19, 21, and 81 variants/Mb with the TSO500 assay. The apparent discordancy of the first sample could be justified as most EQC-contributing labs (24/29) obtained a TMB score <10 demonstrating a good correlation with most analyses. At the RNA level, the SeraSeq FFPE fusion reference RNA was used to assess the accuracy of clinically relevant gene fusions and exon skipping events. As mentioned previously, all 14 gene fusions and 2 exon skipping events were detected in the SeraCare reference sample with a mean of 70 supporting reads (Appendix A). Moreover, the structural aberrations tested for analytical sensitivity and specificity yielded the correct data for all samples (Appendix A). Therefore, the TSO500 assay provides an accuracy of 100% for these samples. 

### 3.6. Tumor Tissue and Specimen Types

Next, we checked our validation data for the availability of several different tumor tissue and specimen types that are typically obtained from the pathology department. Samples were derived from 17 tumor tissue types: lung, colorectal, skin, breast, endometrium, pancreas, ovary, thyroid, brain, GIST, bladder, uterus, neuro-endocrine, prostate, esophagus, stomach, bile, and unknown primary. For tumor tissue specimen-type analysis, we included small (endoscopic) biopts, resections, cytology, endobronchial ultrasound trans-bronchial needle aspiration (EBUS-TBNA), fine needle aspirates (FNAC), and blood. No consistent quantitative or qualitative failures were obtained, and all known variants or rearrangements were correctly detected. Moreover, MSI-H and TMB-H calls were predominantly found in the expected endometrial, ovary, and colorectal tumors. Our data thus indicate that the assay is applicable to a wide range of tumor tissue and specimen types.

### 3.7. Low Input Samples

To assess whether samples with a lower-than-recommended DNA or RNA input yield reliable data, we checked for the presence of such samples in our validation runs. Seven samples had a DNA input lower than the recommended 80 ng due to a too-low DNA concentration and/or a restricted available volume, providing a total input amount of 23 ng to 71 ng. Surprisingly, only one sample did not yield the recommended 150× median coverage, but the known *BRCA2* variant was still correctly detected. Therefore, all seven samples showed full concordance (nine variants and one amplification) and no false-positive variants were found either. At the RNA level, for which 40 ng is the recommended input amount, nine samples had a lower input ranging from 1 ng to 38 ng. We also included a dummy sample with no RNA to check the background and QC values. These data revealed that, except for the dummy sample and one retrospective diagnostic sample, all QC values were within the recommended ranges. The diagnostic sample was repeated but again yielded a similar poor library concentration likely due to insufficient RNA quality. Nevertheless, the repeat analysis correctly identified a HIP::ALK fusion. Our data suggested that lower input amounts can still provide reliable data if the quality-control parameters are met.

### 3.8. TSO500 Screening in Routine

The validated TSO500 assay for detection of a broad range of DNA and RNA variants as well as the biomarkers MSI and TMB was employed to screen diagnostic FFPE samples analyzed from March 2020 to December 2021 in 88 TSO500 runs. Inclusion of the samples was mainly based on a TC > 10%. DNA and RNA were extracted as described previously, and, if possible, 80 ng DNA and 40 ng RNA were used as the starting amounts. Of the 88 runs, 7 were not included because of an insufficient sequencing yield resulting from under- or over-clustering. 

In total, 624 diagnostic DNA and 613 RNA samples of a broad range of tissue types were included. Mean TC was 58% (±24%). For 96 DNA samples, the 80 ng input recommendation was not obtained (range 2 ng to 79 ng), while 118 RNA samples had an input amount <40 ng (range 1 ng to 39 ng). Of the 624 DNA samples, 79 (12.7%) had a median coverage <150× and thus could not be further analyzed, even though reliable tissue-type-specific variants were detected in several of these samples. Notably, 38/79 samples had a starting DNA amount <80 ng (33 samples < 40 ng), suggesting that the insufficient starting DNA amount was the reason for this non-pass feature. Of the 41/79 samples with the recommended 80 ng, 14 had a library concentration <10 ng/µL, likely suggesting low DNA quality. Interestingly, 58 of the 545 samples with a median coverage >150× also had an insufficient DNA starting amount, demonstrating that a minimum of 80 ng DNA should not be taken as a hard cutoff for the assay. In the 545 samples with a sufficient median coverage, at least one SNV or indel (VAF > 5%; coverage ≥50×) was detected in 498 (91.4%) samples, and 203 (37.2%) carried at least one copy number gain (FC ≥ 1.8). For TMB and MSI analysis, 18 (3.3%) samples were MSI-H (>40 MSI sites analyzed with >20% positive), of which 10 were colorectal and 5 were endometrial carcinomas. Six samples were MSI-I (between 10% and 20% positive). A substantial higher number of samples (109; 20%) were TMB-H (variants per Mb ≥ 10). The majority were from endometrial, colorectal, lung, and breast tumors. All 18 MSI-H and 5 out of 6 MSI-I samples were TMB-H. A total of 32 of the 545 DNA samples did not yield any (Likely) Pathogenic SNV, indel, or CNV. These samples were also microsatellite-stable and TMB-low.

Of the 613 RNA samples analyzed, 17 (2.7%) did not reach the threshold of nine million reads. Of the remaining 596 samples, at least one fusion gene was detected in 46 samples, of which 14 had less than 30 supporting reads and thus were regarded as unreliable. Exon skipping events in *AR*, *EGFR*, or *MET* with >100 supporting reads were found in 20 samples.

Finally, of the 587 samples for which DNA and RNA were analyzed, 502 (85.5%) had sufficient DNA and RNA coverage. In 82% (482) of those, at least one SNV, indel, CNV, fusion gene, exon skipping event, or positive biomarker was detected. The total number of SNVs and indels was 1469, which provides a mean of 2.35 per sample.

## 4. Discussion

Here, we validated a comprehensive targeted cancer gene panel for simultaneous detection of SNVs and indels in 523 cancer-related genes, CNVs of 69 genes, gene fusions of 55 driver genes including exon skipping events in *MET* and *EGFR*, and the immunotherapy predictive markers MSI and TMB. The assay was found to fully comply with the different validation performance characteristics for these genetic alterations, allowing reliable diagnostic analysis of solid tumor specimens. This comprehensive genomic profiling assay provides the most extensive molecular information required for cost-efficient patient stratification today. Smaller panels not only have the limitation of the small number of interrogated genes but also the inability to efficiently detect all clinically important copy number variations (CNVs), novel gene fusions, and the TMB biomarker, which precludes the full use of existing or emerging therapeutic strategies. The next step is CPG directly on plasma with, e.g., the TSO500 ctDNA (Illumina) or FoundationOne Liquid CDx (FoundationMedicine) assays.

The processes of DNA/RNA extraction, library preparation, NGS analysis, and reporting can be performed within 4 days, thereby limiting the turn-around time from sample registration to reporting to the clinicians to a maximum of 7 working days, as the test is performed twice a week in our laboratory. Moreover, no matched tumor-normal analysis is required. SNPs, therefore, must be filtered out by their presence in polymorphic databases. As per the ComPerMed guidelines, we are using a minor allele frequency of 0.1% in any ethnic population in GnomAD as the threshold [12]. The method was validated for eight matched DNA and RNA samples, allowing the simultaneous detection of variants at both levels for optimal clinical management. The MSI and TMB scores are predictive for efficient modulation of the response to immunotherapy [14]. Although most homologous recombination repair (HRR) genes are included in the TSO500 panel, HRD analysis is not yet implemented in the assay. Mutational signatures, however, can be retrieved from the data, providing useful information on the tumor [15]. 

Since we have used the accredited TSCA and AmpliSeq for Illumina NGS assays to validate the TSO500 method, we were unable to validate the variants outside the hotspot regions or in genes that were not present in these smaller panels. However, the fact that variants outside these smaller panels were reproducibly detected in the precision studies strongly suggest, in analogy with the validated variants, that these are correct as well. Notably, however, the amplicon coverages obtained with the TSO500 assay were significantly lower than those of the accredited assays due to the incorporated deduplication of the UMI-containing reads. Deduplication removes the PCR-induced duplicate reads, which allows a much more reliable detection of true variants. For that reason, the coverage threshold for TSO500 was set at 50×, compared to 300× for the accredited assays. Interestingly, however, highly similar VAFs were obtained for most variants analyzed with the comparative NGS assays. Therefore, the VAF threshold for SNVs and indels was kept at 5%, which is well above the false positive rate of NGS, certainly in combination with deduplication. These thresholds as well as those for CNVs (≥1.8-fold change), gene fusions (≥30 supporting reads), MSI (>20%), and TMB (≥10 variants/Mb) were based on this validation. Values just below these thresholds were critically judged based on relevant additional information such as the TC, tumor type, and data from other assays such as immunohistochemical analysis. DNA and RNA extracted from fresh (most samples) to up to 2-year-old FFPE samples were used for our validation study. No evidence for deamination was detected based on the absence of a significant increase in the number of variants, with a predominance of C > T; G > A changes. For the diagnostic sample cohort, we mostly used freshly prepared FFPE samples precluding any deamination. Pretreatment of the DNA with Uracil-N-glycosylase (UNG) is therefore not required.

Analytical sensitivity and specificity were maximal for SNVs, indels, CNVs, MSI, and fusions. In the prospective samples, five unknown fusions were detected, which could not be confirmed due to the lack of an orthogonal assay. However, based on the low number of supporting reads and the absence of an inframe fusion with no known driver gene included, three of these fusions were thought to be unrelated to tumorigenesis (Appendix A). It is, however, not straightforward to assess whether these detected rearrangements are false positives or not. 

The TSO500 assay has previously been reported to show a good correlation with whole genome and whole exome sequencing (WES) for TMB analysis [16,17]. Consequently, 20 ng of input DNA was considered sufficient for reliable TMB analysis for samples with a tumor cell percentage ≥ 20% [15]. Although the TMB threshold differs per tumor type [18], we have set this threshold to 10 variants per Mb based on the KEYNOTE-158 study testing TMB in advanced solid tumors [19]. Two of the fifteen samples tested for analytical sensitivity showed a discordant result when compared to the FoundationOne data (Table 3). Both had a TMB value close to the threshold of 10. 

After extensive validation of the different performance characteristics for the diagnostic implementation of this comprehensive genomic profiling panel, we applied the screening to 624 diagnostic FFPE-derived DNA and RNA samples from a wide variety of tumor types. Only samples with a TC ≥ 10% were included, but their inclusion was irrespective of the total input material with a maximum of 80 ng DNA and 40 ng RNA. The median coverage of the 624 diagnostic samples, of which 96 had a starting DNA amount <80 ng, was 477× (±269×). This coverage was substantially higher than the range of 123× to 247× that was reported when using 40 ng DNA as a start for the TSO500 analysis on 108 FFPE samples [15]. Therefore, 80 ng is now recommended for this assay, although lower input amounts can still yield high median coverages. Pre-analytical quality analysis of the samples was not performed because previous analyses showed that most samples passed the test. Moreover, it negatively affects the cost and turn-around time of the assay. However, we expect inferior quality for a large percentage of the samples with maximal DNA input amount that did not yield the 150× median coverage. A manual error in the library preparation was expected for those samples for which the median coverage was below 30×. Automation of the assay should overcome such human errors.

Finally, the TSO500 panel assay has also been successfully validated for the analysis of DNA extracted from myeloid neoplasms, using the same thresholds for coverage (50×) and VAF (5%) [20]. However, the number of samples was limited, and validation was largely performed on reference control samples. Nevertheless, high analytical sensitivity, specificity, precision, and accuracy was reported as well. The clinical value of screening myeloid tumors with such a comprehensive panel is currently still questionable.

## 5. Conclusions

We present one of the largest validation studies of a comprehensive genomic-profiling-targeted panel including more than 500 cancer-related genes for the mutational screening of FFPE samples from a broad range of solid tumors. Next to SNV and indel detection, clinically relevant CNVs, exon skipping events, and fusion genes can be used for therapeutic intervention. Moreover, analysis of the biomarkers MSI and TMB will help to select patients for immunotherapy. Other validation reports investigated a much lower number of genes, less variant types or biomarkers, or a much lower number of retro- or prospective diagnostic samples [21,22,23]. To assess the clinical value of the comprehensive TSO500 analysis compared to small panel screenings, we have set up the real-world BALLETT study (Belgian Approach of Local Laboratory Extensive Tumor Testing; https://clinicaltrials.gov/ct2/show/NCT05058937, accessed on 5 May 2022) with the aim of introducing CGP for precision oncology and to increase treatment options for cancer patients in Belgium. It is expected that an increased progression-free survival rate will be obtained when compared to standard small panel NGS. Hence, our validation set-up can be used as a scaffold for similar initiatives that are currently being set up in other countries.

## Figures and Tables

**Figure 1 cancers-14-02457-f001:**
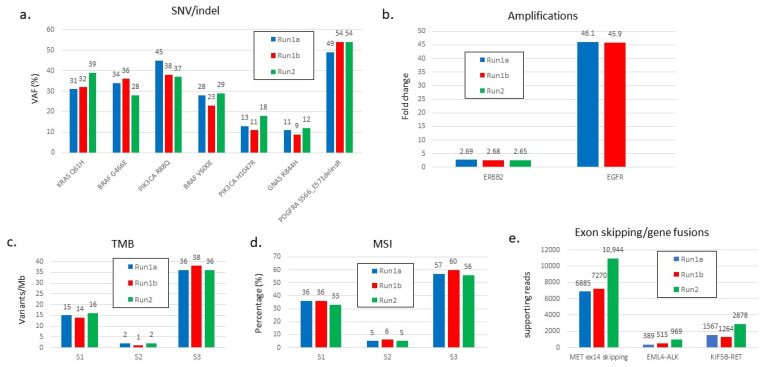
Repeatability and reproducibility of the TSO500 assay for the different types of variants and biomarkers. (**a**) SNVs and indels, (**b**) amplifications, (**c**) TMB, (**d**) MSI, and (**e**) exon skipping and gene fusions events. Three samples were analyzed in the same run (intrarun), indicated as Run1a (blue bars) and Run1b (red bars), as well as in another run (interrun), indicated as Run2 (green bars).

**Table 1 cancers-14-02457-t001:** Overview of the number of variants or biomarkers assessed for each performance characteristic of the TSO500 validation study. The number of samples used are also indicated.

Input	# of Variants at DNA Level					DNA	# of Variants at RNA Level		RNA	Total (DNA + RNA)
Performance Characteristic	SNV	Indel	Amplif	MSI-H	TMB-H	# Smpls	Ex Skipping	Gene Fusion	# Smpls	# Smpls
Precision	27	19	4	4	10	4	1	2	3	7
Sensitivity/Specificity	62	6	6	6	8	90	5	35	80	170
Limit-of-Detection	16	8	8	3	3	14	2	16	5	19
Accuracy	81	6	6	6	8	90	5	35	80	170

#: number; amplif: amplifications; ex: exon; smpls: samples.

**Table 2 cancers-14-02457-t002:** Number of sample types used for the TSO500 validation study.

Sample	Precision		Sensitivity/Specificity/Accuracy		Limit of Detection	
	DNA	RNA	DNA	RNA	DNA	RNA
retrospective	5	3	26	26	13	4
prospective	0	0	48	54	0	0
lung	0	3	19	37	4	2
colorectal	1	0	5	3	3	0
skin	0	0	4	2	0	0
ovary	0	0	7	6	1	0
endometrium	1	0	7	3	0	0
prostate	0	0	1	0	0	1
brain	1	0	2	1	0	0
neuro-endocrine	0	0	3	3	0	0
GIST	1	0	2	0	0	0
breast	1	0	6	1	1	0
pancreas	0	0	2	1	1	0
lymphoid	0	0	0	7	0	0
other ^a^	0	0	6	7	1	0
unknown ^b^	0	0	23	9	2	1
low input ^c^	0	0	7	10	0	0

^a^ including oesophagus, stomach, bile, bladder, and thyroid. ^b^ including reference, EQC, and ring trial samples. ^c^ samples with <80 ng DNA or <40 ng RNA.

**Table 3 cancers-14-02457-t003:** Analytical performance of TMB analysis with the TSO500 assay compared to the FoundationOne assay.

Sample	Foundation One		TSO500	
	TMB Class	# var/Mb	TMB Class	# var/Mb
t1	high	13	high	17
t2	low	1	low	1
t3	low	3	low	8
t4	low	6	high	10
t5	high	13	high	14
t6	high	10	low	9
t7	low	4	low	2
t8	low	0	low	3
t9	high	30	high	39
t10	high	14	high	17
t11	low	8	low	5
t12	low	4	low	4
t13	high	14	high	11
t14	high	10	high	11
t15	high	26	high	34

#: number; both discordant calls are highlighted in grey.

## Data Availability

Data are contained within the article or Appendix A. Relevant parts of these patient-derived data can be obtained upon request.

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
