# Peer review of "Diagnostic Validation of a Comprehensive Targeted Panel for Broad Mutational and Biomarker Analysis in Solid Tumors"

_cancers, 2022, doi:10.3390/cancers14102457_

Round 1

Reviewer 1 Report

The authors of manuscript no.: 1701391 provide results of comprehensive validation of the TruSight Oncology (TSO500) assay, using more than 500 cancer-related genes, designed for detection of SNVs, small indels, CNVs, gene fusions, MSI, and TMB. The first part of the study demonstrated analytical sensitivity and specificity of at least 99% for SNV, indels, CNV, MSI, and gene rearrangements. Then 624 DNA and 613 RNA clinical samples from FFPE blocks representing several different tumors and specimen types have been used for routine screening and manifesting the utility of the assay. The study is very important for precision oncology and further translation of molecular data to clinically useful biomarkers and treatment decisions. The manuscript is quite clearly written and technically sound. There are a few points that should be considered in the revised version:

1/ RNA from FFPE was used. What was the quality (RIN or another measure) of these samples? How was it controlled? How old were FFPE samples? It should be stated in Methods. Was there a correlation between mutation numbers or load with the age of specimens? Would the TSO500 assay work also with real archival FFPE samples older than 5-10 years or so? Overall, data on base changes should be presented to document the “no evidence for deamination” in DNA from FFPE. How many freshly prepared and how many older (and how long time fixed) samples were used?

2/ The study design does not seem straightforward when reading only the text. Some kind of study flow chart with main decision steps and outputs would be more informative and easier to deal with. Also, an overview of sample types (cancer type, specimen type, prospective/retrospective, etc.) with their numbers used in each study phase would be useful as this information is now just scattered throughout the text and not easy to follow.

3/ I have one question for the general discussion. Is it possible to use the TSO500 assay for the analysis of cell-free samples or exosome specimens?

4/ Anonymized sequencing data should be made available in a public repository such as SRA (NCBI) to allow for replication of the study or independent reassessment. If that is not possible, an explanation should be provided (section Data Availability Statement).

Minor comments:

  • Page 3, line 114: Please explain what you mean by (Qubit) “RNA HR assay kit”? Should not it be HS?
  • Page 4, lines 154-155: How exactly did you process data of tumor samples without matched non-malignant reference (tumor-normal)? Could you provide more details?
  • Page 4, line161: please specify the GnomAD version.
  • Page 4, line 178: specify problematic samples – DNA or RNA and how many?
  • Page 5, line 198, “(S1 and S3)” should be harmonized with Tab 2 where samples are shown as s1-s15 which is inconsistent.
  • Page 5, line 209, Figure 1e: I would not call the number “comparable”, which implies that they are very similar, which they are not. Run 2 had many more reads than the other runs.
  • Figure 1 in general: please point to specific parts of a figure in the text where appropriate instead of just Fig. 1.
  • Page 7, line 287: Please explain the abbreviation “EQC”.
  • English language polishing may be advised in some instances.

Author Response

Please find our comments attached in the file 'Response to Reviewer 1'.

Reviewer 2 Report

Froyen et al described the validation and implementation of the hybrid capture based comprehensive TSO500 assay to detect SNVs, indels, certain gene fusions, and MSI & TMB.  While the topic is of interest to the field, there are some issues that need to be addressed before publication.

  1. the authors used the FFPE samples and claim detecting >99% for all variant types. FFPE samples are known to cause sequencing artifacts and this at least needs to be discussed in the paper.
  2. next generation sequencing has 0.1-1% false positive rate per bp. How do authors distinguish false positives with rare true positives. Additional validation is probably needed.
  3. it might be beyond the scope of this paper experimentally, but the authors should discuss error-corrected duplex sequencing approaches that allow detection of true rare variants.
  4. Figure legend for figure 1 not very clear. what's run 1a, 1b and 2?
  5. Figure 1e, green bar seems to be very different from run 1a, 1b. any explanation for this?
  6. It is not clear how the larger panel can benefit the diagnosis/treatment of cancer, please discuss. 

Author Response

See the attached file 'Response to Reviewer 2'

Reviewer 3 Report

In this study the Authors validated a comprehensive targeted cancer gene panel for simultaneous detection of SNVs and indels in 523 cancer-related genes, CNVs of 69 genes, gene fusions of 55 driver genes including exon skipping events in MET and EGFR, and the immunotherapy predictive markers MSI and TMB.

The Authors demonstrated that this assay is a reliable tool for analysis of genomic alterations in tumor specimens.

Author Response

We would like to thank the expert for reviewing our paper and for the very positive response.

Round 2

Reviewer 2 Report

authors have addressed my concern.